# The Highly Sensitive Refractive Index Sensing of Seawater Based on a Large Lateral Offset Mach–Zehnder Interferometer

**DOI:** 10.3390/s24123887

**Published:** 2024-06-15

**Authors:** Jingwen Zhou, Yue Sun, Haodong Liu, Haibin Li, Yuye Wang, Junfeng Jiang, Degang Xu, Jianquan Yao

**Affiliations:** 1School of Marine Science and Technology, Tianjin University, Tianjin 300072, China; zhoujingw2009@163.com; 2School of Precision Instruments and Optoelectronics Engineering, Tianjin University, Tianjin 300072, China; sunyue102695@outlook.com (Y.S.); haodong_liu@tju.edu.cn (H.L.); haibin_li@tju.edu.cn (H.L.); jiangjfjxu@tju.edu.cn (J.J.); jqyao@tju.edu.cn (J.Y.)

**Keywords:** refractive index sensor, open-cavity Mach–Zehnder interferometer, high sensitivity

## Abstract

A novel fiber sensor for the refractive index sensing of seawater based on a Mach–Zehnder interferometer has been demonstrated. The sensor consisted of a single-mode fiber (SMF)–no-core fiber (NCF)–single-mode fiber structure (shortened to an SNS structure) with a large lateral offset spliced between the two sections of a multimode fiber (MMF). Optimization studies of the multimode fiber length, offset SNS length, and vertical axial offset distance were performed to improve the coupling efficiency of interference light and achieve the best extinction ratio. In the experiment, a large lateral offset sensor was prepared to detect the refractive index of various ratios of saltwater, which were used to simulate seawater environments. The sensor’s sensitivity was up to −13,703.63 nm/RIU and −13,160 nm/RIU in the refractive index range of 1.3370 to 1.3410 based on the shift of the interference spectrum. Moreover, the sensor showed a good linear response and high stability, with an RSD of only 0.0089% for the trough of the interference in air over 1 h.

## 1. Introduction

The ocean covers 75% of the surface of the Earth and is widely regarded as the birthplace of life [1]. The exploitation of the oceans has significant ramifications for a number of large-scale phenomena, including climate change, the sustainable development of resources, and national security [2]. Changes in environmental conditions can have a significant impact on the distribution of fish, seaweed growth, marine engineering, and other organisms [3,4]. Seawater salinity is a prominent factor in marine environmental monitoring as it is the parameter of conductivity [5,6]. Seawater conductivity is usually obtained from research vessels [7], underwater gliders [8], underwater vehicles [9], or other similar devices. However, this method has some drawbacks, including its susceptibility to electromagnetic interference and the vehicles’ large sizes [10]. Comparatively, the optical fiber sensing method has the advantages of anti-electromagnetic interference, a small size, fast responses, high precision, and a compact structure [11]. It has been widely applied to obtain the salinity of seawater via refractive index (RI) sensing [12].

Until now, several fiber sensing methods have been proposed to measure the RI of seawater, such as long-period fiber grating (LPFG), micro-structured fiber, and fiber interferometer methods [13]. The long-period fiber grating RI measurement method can achieve a high sensitivity in RI sensing. On the one hand, double peak resonance is achieved by designing the period of the LPFG and its cladding mode using a phase matching curve (PMC) to find the dispersion turning point (DTP). On the other hand, to improve the RI sensitivity of long-period fiber grating (LPFG)-based sensors, a mode transition (MT) effect has been proposed. This is usually achieved by using coated and double-cladded fibers [14]. DTP methods only increase the sensitivity to an order of magnitude of a thousand, which is less sensitive in seawater RI sensing [15,16,17]. Moreover, the repeatability and stability of the sensor when using the MT method is easily affected due to the need for special coating materials [14,18]. Currently, with the development of micro-structured fibers, many micro-structured fibers have been studied for RI sensing. Micro-structured fibers are typically etched and tapered to enhance the RI sensitivity of a sensor. These sensors are typically susceptible to damage and have a short lifespan due to the reduced diameter of their fibers [19].

The interferometers used for RI sensing can be classified into Fabry–Perot interferometers (FPIs) and Mach–Zehnder interferometers (MZIs), depending on their structure. The main focus of FPI sensors is the fabrication of FP cavities. These cavities are usually produced through chemical deposition methods or by combining fiber structures with different modes [20]. Compared to the FPI sensors, the MZI sensors are more stable because they do not require coatings. In particular, in 2019, a lateral offset MZI sensor was proposed and the effect of different vertical axial offset distances (ranging from 6 to 40 μm) on the RI sensitivity of the sensor was explored, achieving a seawater RI sensitivity of 123.40 nm/RIU [21]. Moreover, the inclusion of a multimode fiber (MMF) into its large offset structure can improve the offset tolerance of this sensor [22]. The vertical axial offset distance of the sensor was increased to 62.5 μm, creating an open-cavity structure, which achieved a high RI sensitivity of −2953.444 nm/RIU in the RI range of 1.33302–1.33402. Recently, an offset-based MZI sensor for the measurement of the refractive index of seawater has been reported. This sensor is capable of measuring the refractive index of seawater over a range of 1.333–1.334, where its sensitivity reached to 11,000 nm/RIU, with an uncertainty of 0.00001 RIU [23]. The incorporation of a fluorine-doped fiber, which exhibits a lower refractive index than the cladding, has been demonstrated to enhance sensitivity by approximately 10% compared to the sensitivity of a conventional single-mode fiber [24]. Furthermore, the method of online combinations has been used to increase the number of interferences, thereby enhancing the sensitivity of the process [25].

In this paper, an open-cavity MZI fiber-optic sensor with a lateral offset SMF-NCF-SMF (SNS) structure incorporated into its MMF (SM-OSNS) has been proposed, improving its RI sensitivity. Optimization studies of its multimode fiber length, offset SNS length, and vertical axial offset distance were performed to improve the coupling efficiency of the interference light and achieve the best extinction ratio (ER). Moreover, the sensor allowed for an increased tolerance of the fiber core offset and reduced the difficulty of fabrication. In the experiment, a large lateral offset sensor was prepared to detect the RI of various salinities of saltwater, which were used to simulate seawater environments. The sensing sensitivity was up to −13,703.63 nm/RIU and −13,160 nm/RIU within the RI range of 1.3370–1.3410. Moreover, the sensor exhibited a strong linear relationship and high stability, with minimal fluctuations in its transmission spectrum in air over 1 h. It could be a valuable tool for ocean exploration.

## 2. Theoretical Analysis

Figure 1 shows the structure of the sensor, which comprised two sections of multimode fiber (MMF1, MMF2) connected to a section of SNS. The lengths of the individual structures are indicated in Figure 1, where $L_{offset}$ is the vertical axial offset distance and L indicates the physical length of the offset SNS structure. The MMF1 acted as a beam expander, allowing light transmission and exciting higher-order modes to increase the coupling area. The lateral offset structure comprised a small section of NCF connected between two sections of SMF (SMF1, SMF2). The core diameters of the SMF and MMF were 8.3 μm and 105 μm, respectively, and the cladding diameters were both 125 μm. The cladding of the no-core fiber diameter was also 125 μm. It created an open cavity between the offset SNS structure and the MMFs, allowing the fiber to transmit the light field while in direct contact with the sensing environment, thereby increasing the sensitivity of the sensor. The coupling area between the input light and output light was increased by the addition of the MMF with the open cavities.

The transmitted light field of the SM-OSNS sensor in air was simulated using the beam propagation method (BPM), as shown in Figure 2a. Figure 2b shows the simulated transverse optical fields of each section. It was shown that the light field’s energy was concentrated in the core part of the lead-in SMF and that the diameter of the transverse optical field increased significantly after entering the MMF1. At the first fusion splice point, the light was split into two beams due to a large difference in their RIs. One beam propagated in the surrounding environment as the sensing arm, while the other propagated in the SNS structure as the reference arm. The beams in the sensing arm and the reference arm were recombined in the MMF2, creating Mach–Zehnder interference. The optical intensity of the interference output can be expressed as
(1)$$I\left(\lambda \right)=I_{1}+I_{2}+2\sqrt{I_{1}I_{2}}\cos\left(\Delta \varphi \right)$$
where the optical intensities of the sensing arm and reference arm are represented by $I_{1}$ and $I_{2}$ respectively. The phase difference Δφ can be expressed by the following equation:(2)$$\Delta \varphi =\frac{2\pi \Delta n_{eff}L}{\lambda }$$
where $\Delta n_{eff}$ represents the effective RI difference between the two interfering arms, λ represents the wavelength of the input broadband light in the vacuum, $n_{e}$ is the RI of the external environment. $\Delta n_{eff}L=(n_{e}-n_{SMF1})L_{SMF1}+(n_{e}-n_{NCF})L_{NCF}+(n_{e}-n_{SMF2})L_{SMF2}$. Here, $n_{SMF1}$, $n_{NCF}$, and $n_{SMF2}$ are the effective RIs of the light propagating in SMF1, NCF, and SMF2. The phase difference Δφ should be negative because the refractive index of seawater is lower than that of the optical fiber. Due to the varying wavelengths of light in the broadband spectrum, there is a phase difference that results in changes to the intensity of the interference light. When $\varphi_{0}=\left(2m+1\right)\pi$, where m is an integer, the interference light intensity exhibits as a trough. The corresponding resonance wavelength $\lambda_{m}$ is
(3)$$\lambda_{m}=\frac{2\pi \Delta n_{eff}L}{\Delta \varphi }=\frac{2\pi \Delta n_{eff}L}{(2m+1)\pi }=\frac{2\Delta n_{eff}L}{\left(2m+1\right)}$$

The free spectral range (FSR) is defined as
(4)$$FSR=\lambda_{m+1}-\lambda_{m}=\frac{\lambda_{m}\lambda_{m+1}}{\Delta n_{eff}L}$$

The FSR is a crucial performance parameter for RI measurements. Considering that the FSR is inversely proportional to both the length of the sensing structure and its effective RI difference, a broadening of the transmission spectrum appears when the ambient RI difference increases. Thus, a wider range of RI can be measured as the FSR of the interference fringe is increased.

In traditional fiber sensors, the RI sensitivity is limited by the evanescent field on the fiber surface. In the case of open-cavity structures, the surrounding environment is considered as one part of the waveguide, and it interacts with the light directly. Thus, an open-cavity MZI structure is highly sensitive to even slight changes in external RI, resulting in a noticeable shift in the interference trough of the transmission spectrum. By differentiating Equation (2), the sensing sensitivity of the SM-OSNS sensor can be expressed as
(5)$$\frac{d\lambda_{m}}{dn_{e}}=\frac{2\pi }{\Delta \varphi }\left[L+n_{e}\frac{dL}{dn_{e}}-\left(\frac{dn_{SMF1}}{dn_{e}}L_{SMF1}+n_{SMF1}\frac{dL_{SMF1}}{dn_{e}}+\frac{dn_{NCF}}{dn_{e}}L_{NCF}+n_{NCF}\frac{dL_{NCF}}{dn_{e}}+\frac{dn_{SMF2}}{dn_{e}}L_{SMF2}+n_{SMF2}\frac{dL_{SMF2}}{dn_{e}}\right)\right]$$

Because the length in sensing region is changed a little during sensing, it can be negligible, as dL≈0, and Equation (5) can be simplified as follows:(6)$$\frac{d\lambda_{m}}{dn_{e}}=\frac{2\pi }{\Delta \varphi }\left[L-\left(\frac{dn_{SMF1}}{dn_{e}}L_{SMF1}+\frac{dn_{NCF}}{dn_{e}}L_{NCF}+\frac{dn_{SMF2}}{dn_{e}}L_{SMF2}\right)\right]$$

Actually, the RI change of the external environment is much larger than that of the fiber cladding. Thus, $n_{SMF1}$ and $n_{SMF2}$ in Equation (6) can be approximated as the RIs of the cladding of the SMF and $n_{NCF}$ can be approximated as the RI of the NCF. According to Equation (6), to enhance the sensing sensitivity of the sensor, it is necessary to decrease the absolute values of Δφ and $\frac{dn_{SMF1}}{dn_{e}}L_{SMF1}+\frac{dn_{NCF}}{dn_{e}}L_{NCF}+\frac{dn_{SMF2}}{dn_{e}}L_{SMF2}$. When the NCF is added, the RI change from the offset SNS structure to the external environment decreases. This can result in a decrease in the absolute value of Δφ and an increase in the numerator value of the sensing sensitivity. In other words, the adding of an NCF can effectively improve the RI sensitivity of the sensor. In addition, the extinction ratio (ER) is used to evaluate the spectral quality of the interference spectrum, and this can be expressed as [22]:(7)$$ER=10\cdot {\mathrm{lg}}_{10}\frac{I_{\max}}{I_{\min}}$$

$I_{\max}$ and $I_{\min}$ represent the peak and the trough of the wavelength of the interfering light. As the ER is increased, the performance of the sensor will improve.

## 3. Design of the Sensor

### 3.1. Optimization of the Sensor

To enhance the sensitivity of the fiber sensor, the parameters of the sensor length and the vertical axial offset distance were optimized. Firstly, the optimum lengths of the SMFs at both ends of the offset structure were calculated. Figure 3a showed the FSR and ER under different lengths of SMFs, where the vertical axial offset distance was set as 62.5 μm, the length of the MMFs was 500 μm, and the length of the NCF in the offset structure was 200 μm.

It is seen that the FSR of the fiber sensor gradually decreased as the length of the SMF on each side of the offset structure was increased. When the length of the SMFs was increased from 50 μm to 250 μm, the FSR of the sensor decreased from 74.8 nm to 28.7 nm. Additionally, it was observed that the ER of the sensor reached its maximum when the lengths of SMF1 and SMF2 were 100 μm. Figure 3b shows the relationship between SMF length and SM-OSNS sensor sensitivity. It can be clarified that the sensitivity of the sensing structure decreases as the length of SMF1 and SMF2 increases from 50 μm to 250 μm. Considering the spectrum range of the laser source, the SMF lengths on both sides were set as 100 μm in the following experiment.

Furthermore, the RI sensitivity under different lengths of NCF was calculated with the SMF = 100 μm, as shown in Figure 3c. The absolute value of RI sensitivity was increased from 13,209 nm/RIU to 14,507 nm/RIU as the NCF length was increased from 100 μm to 400 μm, but it was decreased to 14,018 nm/RIU when the NCF length was increased from 400 μm to 500 μm. Therefore, the SM-OSNS sensor had the highest sensitivity when the NCF was 400 μm. However, it is known from Equation (4) that the FSR of the sensor decreases as the offset length is increased. Therefore, the length of the NCF in the offset structure was finally set to 200 μm to ensure the high sensitivity of the sensor and a large RI measurement range.

The type of fiber fused to both ends of the offset SNS structure affected the transmission spectral loss and ER of the sensor significantly. Figure 4a shows the transmission spectra of the multimode fiber (SM-OSNS) and single mode fiber (SSNSS) at the two ends of the offset SNS structure. The total offset length and the vertical axial offset distance were set as 400 μm and 62.5 μm for the simulated calculation, respectively. In the offset SNS structure, the lengths of the SMFs at both ends were 100 μm, and the length of the NCF was 200 μm. The lengths of the MMFs in the SM-OSNS sensor structure were both 500 μm. Compared with the SSNSS, it is shown that the loss of the sensor was reduced to −16.55 dB after the addition of MMFs. The ER of the sensor also becomes twice as high as before. Therefore, a high ER sensor can be achieved by selecting an appropriate MMF length. To optimize the lengths of MMF1 and MMF2, the transmission spectra of MMF sensors with different lengths were simulated in air with MMF1 = MMF2, as shown in Figure 4b. The parameter values of the vertical axial offset distance and the offset SNS length were same as those used in Figure 4a. It can be seen that the ER of the transmission spectrum in air reached a maximum value of 44.93 dB with MMFs of 500 μm. Therefore, the lengths of the MMF1 and MMF2 were selected as 500 μm.

The vertical axial offset distance is a crucial parameter for offset fiber sensors which affects the ER of the transmission spectrum of the sensor directly. Figure 5a,b show the transmission spectra in seawater for different vertical axial offset distances when the MMFs and SMFs were used as the two ends, respectively. The offset SNS length was 400 µm. It can be seen from Figure 5a that the ER of the transmission spectrum was increased from 1.29 dB to 45.55 dB and then decreased to 1.05 dB as the vertical axial offset distance was increased from 50 μm to 75 μm, with the largest ER obtained at a vertical axial offset distance of 62.5 μm. Thus, the vertical axial offset distance of 62.5 μm was selected in our experiment. The simulation results showed that the ERs of the transmission spectra were larger than 5 dB when the vertical axial offset distance was in the range of 55 to 70 μm, which ensured that the fiber sensor was compatible with most wavelength demodulation systems. Comparatively, the transmission spectra of the SSNSS structure with vertical axial offset distances of 60–65 μm were also calculated, as presented in Figure 5b. This illustrated that the ER of the SSNSS was greater than 5 dB when the vertical axial offset distance was only in the range of 61 to 64 μm, as its fusion tolerance was much smaller than that of the SM-OSNS sensor. Thus, the SM-OSNS structure showed obvious advantages compared to the SSNSS structure. The addition of an MMF increased the overlap area between the fiber core and the offset SNS structure, which improved the offset tolerance of the sensor and reduced the requirement for offset accuracy in the fabrication process.

### 3.2. Simulation of Sensor Transmission Spectrum

Based on the optimized parameters above, the transmission spectra of the SM-OSNS sensor in both air and seawater were simulated, as shown in the inset of Figure 6a. The FSRs were 13.3 nm and 53.3 nm in air and seawater, and the ERs were 14.1 dB and 43.9 dB in air and water, respectively. The RI of the seawater ranged from 1.3382 to 1.3408 [26]. Figure 6 depicted the drift of the transmission spectra of the sensor in solutions with different refractive indices, where the RI ranges from 1.3370–1.3410 with a gradient of 0.0005. The transmission spectra showed a blueshift phenomenon as the RI of the solution increased, which can be seen in Figure 6a. Figure 6b shows the linear relationship between the resonance wavelength and the RI. The RI sensing sensitivities of the two resonance dips (labeled as dip A and dip B, respectively) were calculated to be −13,387 nm/RIU and −13,849 nm/RIU, with a linearity of 0.999 and 0.999.

## 4. Experiment and Discussion

### 4.1. Fabrication of Fiber-Optic Sensor

Figure 7a–c depict the experimental structure preparation process of the SM-OSNS sensor. The main operations included fiber cutting, fusion splicing, and offset fusion splicing. The SMF and MMF were automatically fused using a fiber-optic fusion splicer (LDS 2.5, 3SAE). Subsequently, the MMF was cut using an ultrasonic fiber cutter with precise length control and retained at a length of 500 μm. The SNS structure at one end was fusion spliced with the SMF-MMF, with an offset distance of 62.5 μm in the vertical axial direction and no offset in the parallel axial direction. The current was dynamically adjusted between 600 and 750 bits during the fusion splicing process to prepare a sensor with higher mechanical strength. After the fusion splice stage, the length of the offset SNS structure was cut and retained at 400 μm, and the above operation was repeated to splice the other side. Figure 7d depicts our prepared SM-OSNS sensor with a vertical axial offset distance of 62.5 μm; MMF1 and MMF2 lengths of 490 μm and 495 μm; and SMF1, NCF, and SMF2 lengths of 92 μm, 170 μm, and 108 μm, respectively. It was observed that the actual parameters were slightly different from the optimized theoretical parameters above due to fabrication errors.

### 4.2. Experimental Procedures

Figure 8 displays the experimental setup for measuring the RI of our prepared seawater. The setup included a broadband light source (ASE-CL-30-M, Max Ray Photonics, Hefei, China), SM-OSNS sensor and liquids with various RIs, as well as a spectrum analyzer (OSA, YOKOGAWA AQ6370D, Musashino, Japan). The ASE light source emitted light with a wavelength range of 1530–1600 nm and a power of 27 mW. The light carried information about the external environment after passing through the sensor, which was then received and analyzed by the OSA, allowing the RI of the external environment to be detected.

In the experiment, seawater was prepared, using a NaCl solution, with an RI range of 1.3370–1.3410 and a gradient of 0.0005. The RI of the solution was measured using an Abbe refractometer (WAY-2WAJ, LICHEN, Shanghai, China). We sequentially immersed the SM-OSNS sensor into the prepared seawater solutions. The transmission spectra were measured to determine the RI sensing characteristics of this sensor. All experiments were conducted at 22 °C. To guarantee the stability of the sensor’s structure, the sensor was fixed on a microscope slide with UV glue for each measurement. Then, the sensor was rinsed with deionized water and dried until the original spectrum was restored in air. Thus, the repeated use of the sensor can be well guaranteed.

### 4.3. Results and Discussions

Figure 9 shows the transmission spectra of the SM-OSNS in air and seawater. The FSRs of the fabricated sensor were 12.1 nm and 57.5 nm in air and seawater, respectively, which were consistent with the simulated spectra in the inset of Figure 6a. This means that the RI sensing of seawater with an SM-OSNS sensor is feasible. It should be mentioned here that the loss and ER of the transmission spectrum in air were −26.57 dB and 20.41 dB, respectively, which were greater than the simulated results. This can be attributed to the extra fusion loss during the fiber fusion splicing process and the difference between the actual and simulated structural parameters during fabrication.

Figure 10a displays the transmission spectra of the sensor in solutions with various RIs. The values of the ER of the transmission spectra for different RIs were approximately −12 dB. The results indicated that the increase in the RI caused the blueshift in the transmission spectra; namely, the resonance dips (labeled as dip A and dip B, respectively) shifted towards shorter wavelengths. Figure 10b illustrates the relationship between the resonant wavelength and RI for dip A and dip B. The sensor exhibited RI sensitivities of −13,703.63 nm/RIU and −13,160 nm/RIU within the RI range of 1.3370–1.3410, with a linearity of 0.990 and 0.997, respectively. To ensure measurement stability, the SM-OSNS sensor was kept static in air for one hour. The transmission spectra were collected every ten minutes. As shown in Figure 11a, the resonance wavelength of the transmission spectrum in air remained relatively constant over time. Moreover, the drift of the three dips in the transmission spectrum in Figure 11a were monitored separately over 1 h, as shown in Figure 11b. The relative standard deviations (RSDs) for the troughs, of 1535 nm (dip1), 1559 nm (dip2), and 1584 nm (dip3), were only 0.0102%, 0.0091%, and 0.0089% in air over 1 h, respectively. This fluctuation may be caused by external environmental factors, such as temperature and pressure. Therefore, the packaging of the SM-OSNS sensor is necessary for its real application.

Table 1 presents the performance of the proposed SM-OSNS sensor and other sensors. It can be observed that the proposed sensor exhibits notable advantages in seawater refractive index sensing with regard to the specific requirements for refractive index sensitivity, sensor stability, detection range, and fusion accuracy. The SM-OSNS sensor had a sensitivity of −13,703 nm/RIU in the RI range of 1.3370–1.3410, which showed the significant advantage of its sensing sensitivity. Its practical application must consider the impact of seawater washing, chemical corrosion, and biological attachment, as well as other factors in the marine environment. Therefore, a high-stability packaging of the SM-OSNS sensor should be adopted, such as capillary packaging [22]. Moreover, a physical compensation mechanism should be considered to correct the test results.

## 5. Conclusions

In conclusion, an SM-OSNS fiber sensor has been demonstrated for the RI sensing of seawater. The sensor was composed of an SNS structure embedded between two sections of MMF. Optimization studies were conducted on the length of the multimode fiber, the length of the offset SNS, and the vertical axial offset distance to improve the coupling efficiency of the interference light and obtain the best ER. The experimental results indicated that the SM-OSNS sensor exhibited a good linear response to seawater RIs in the range of 1.3370–1.3410, with a high sensitivity of up to −13,703.63 nm/RIU. This sensor showed a wide range of applications in the measurement of the RI of seawater.

## Figures and Tables

**Figure 1 sensors-24-03887-f001:**
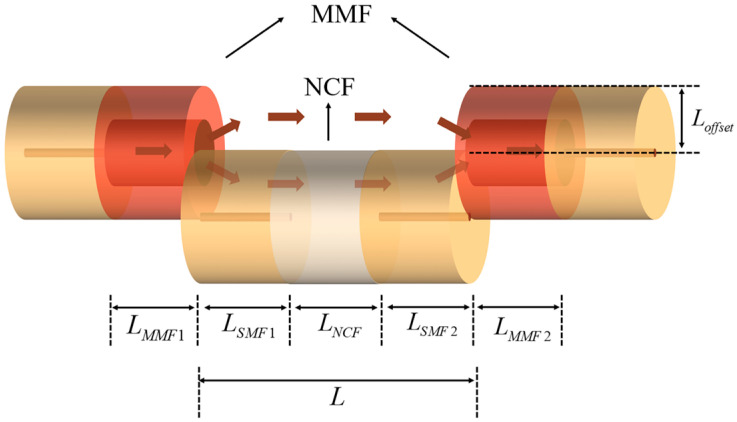
Schematic diagram of the SM-OSNS sensor.

**Figure 2 sensors-24-03887-f002:**
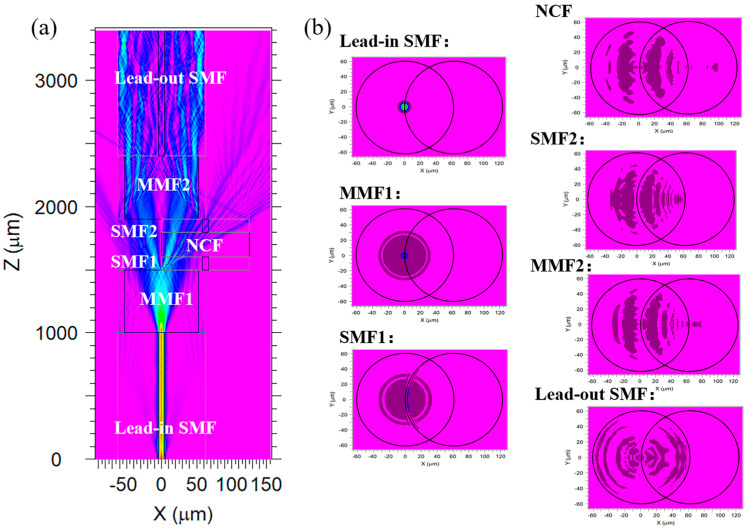
(**a**) Simulation of optical field distribution of the SM-OSNS; (**b**) contour maps of the transverse optical fields of each section.

**Figure 3 sensors-24-03887-f003:**
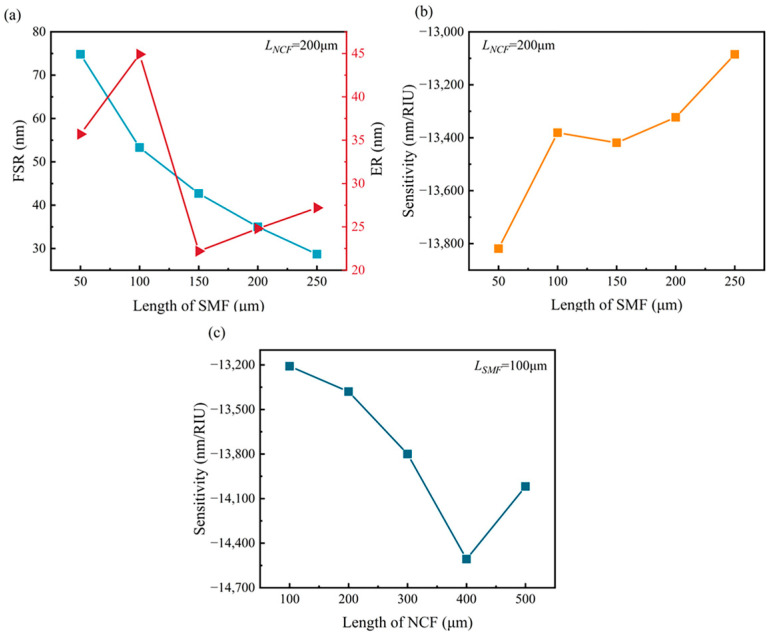
(**a**) FSR and ER with different lengths of SMF for NCF = 200 μm; (**b**) RI sensitivity with different lengths of SMF for NCF = 200 μm; (**c**) RI sensitivity with different lengths of NCF for SMF = 100 μm.

**Figure 4 sensors-24-03887-f004:**
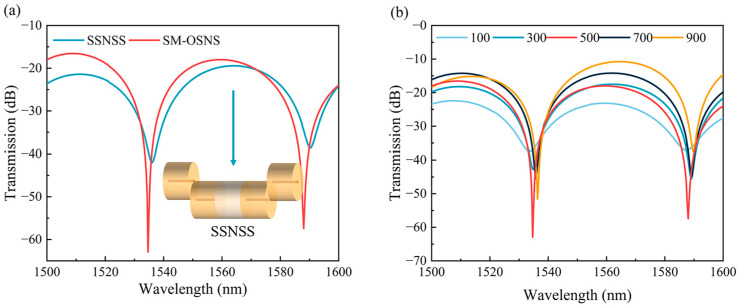
(**a**) Transmission spectra of different structures in seawater; (**b**) simulation of transmission spectra with different MMF lengths.

**Figure 5 sensors-24-03887-f005:**
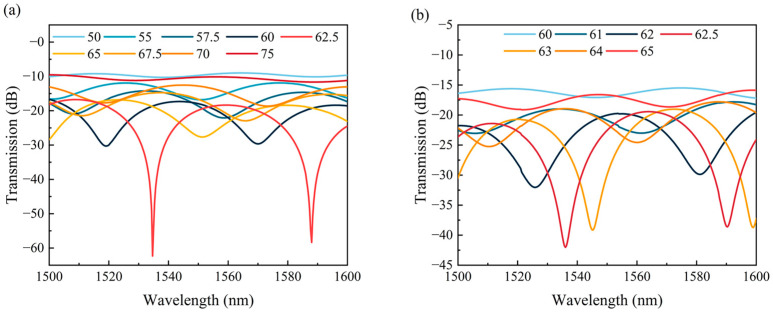
Relationship between sensor transmission spectra and different vertical axial offset distances: (**a**) SM-OSNS structure; (**b**) SSNSS structure.

**Figure 6 sensors-24-03887-f006:**
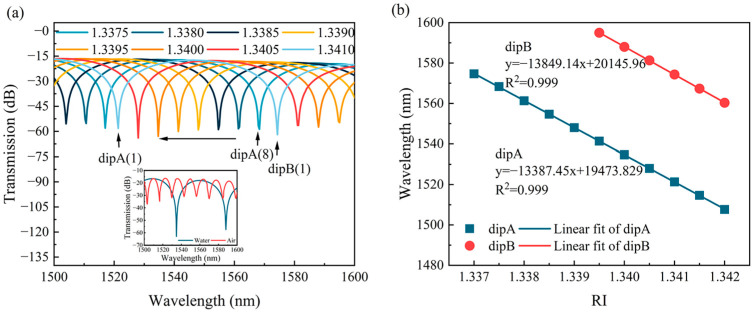
Simulated spectra of RIs in various environments: (**a**) transmission spectra of sensors with different RIs (inset was in water and air); (**b**) RI linear fitting results.

**Figure 7 sensors-24-03887-f007:**
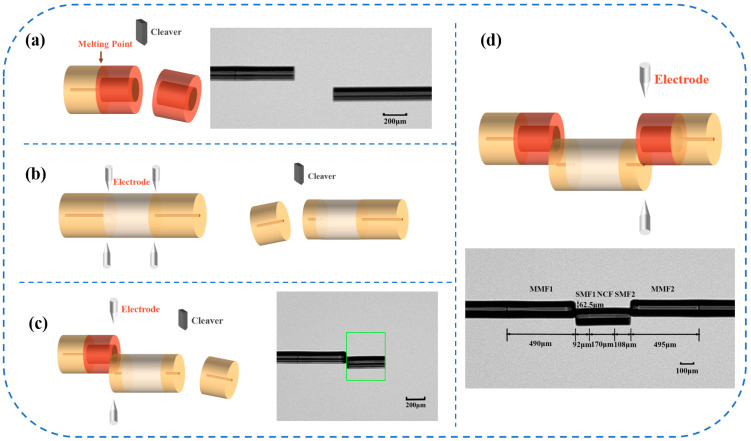
(**a**–**c**) Schematic of various stages of fusion splicing; (**d**) schematic of sensor structure.

**Figure 8 sensors-24-03887-f008:**
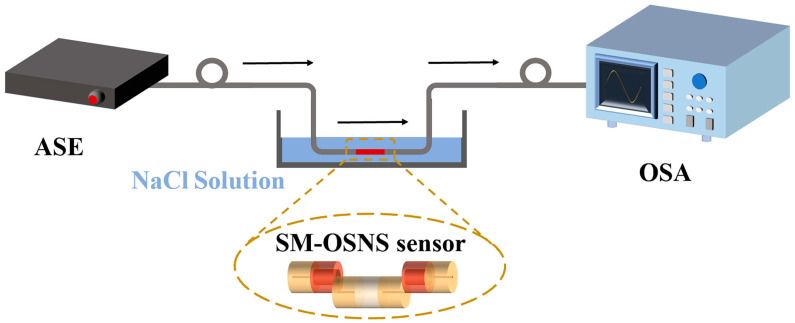
Experimental setup for RI measurement based on SM-OSNS sensor.

**Figure 9 sensors-24-03887-f009:**
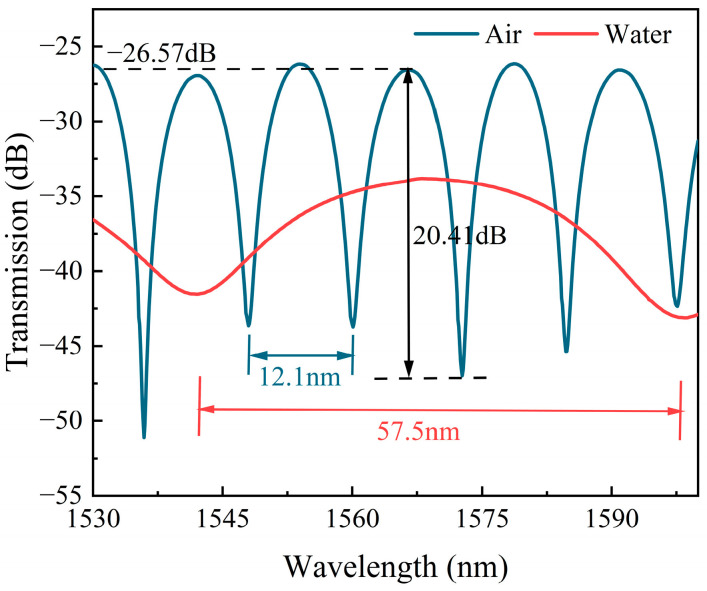
Transmission spectra of SM-OSNS sensor in air and seawater.

**Figure 10 sensors-24-03887-f010:**
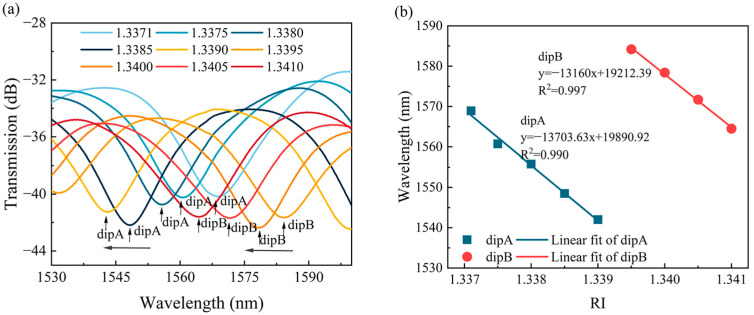
Experimental spectra of RIs in different environments: (**a**) RI response curve of the sensor; (**b**) linear fitting of RI responses.

**Figure 11 sensors-24-03887-f011:**
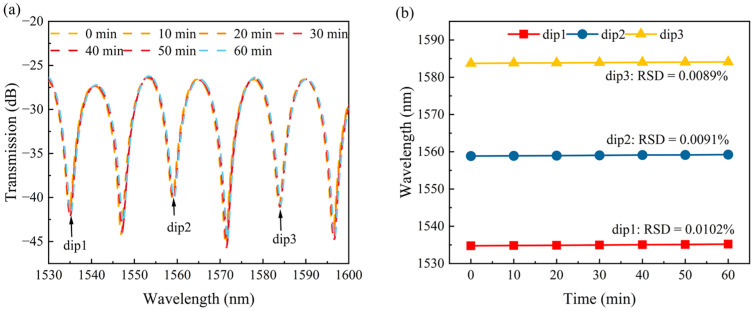
(**a**) The spectrum variation of the sensor over 1 h in air; (**b**) stability lines of different dips.

**Table 1 sensors-24-03887-t001:** Comparison of sensor performance between previous sensors and the proposed sensor.

Application	Sensor	Advantageous Interferometer Type	Measurement Range	Sensitivity	Ref.
RI sensor	Tapered seven-core fiber	MZI	1.3330–1.3451	1435.76 nm/RIU	[27]
Salinity sensor	SMF-MMF-etched DSHF-MMF-SMF	MZI	0–40‰ (1.3313–1.3395)	−2 nm/‰ (−10,872 nm/RIU)	[28]
Salinity sensor	SMF-OFFSET-SMF-SMF	MZI	20–40‰	−2.4473 nm/‰ (~−15,000 nm/RIU)	[29]
RI sensor	Microfiber-Assisted U-Shape Cavity	MZI	1.3197–1.3250, 1.3434–1.3475	−8449 nm/RIU, −13,245 nm/RIU	[30]
RI sensor	SMF-OFFSET-SMF-SMF	MZI	1.3328–1.3398	13,936 nm/RIU	[31]
RI sensor	Superimposed coated LPG-FBG	LPFG-FBG	1.3300–1.3420	2326.7 nm/RIU	[32]
RI sensor	PCF-MMI	MMI-PCF	1.3330–1.3775	342.78 nm/RIU	[33]
RI sensor	SM-OSNS	MZI	1.3370–1.3410	−13,703 nm/RIU	This paper

## Data Availability

Data is contained within the article.

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
