# Peer review of "The Highly Sensitive Refractive Index Sensing of Seawater Based on a Large Lateral Offset Mach–Zehnder Interferometer"

_sensors, 2024, doi:10.3390/s24123887_

Round 1

Reviewer 1 Report

Comments and Suggestions for Authors

This manuscript proposed a novel fiber sensor for the refractive index sensing of seawater based on a Mach-Zehnder interferometer has been demonstrated. The sensor consisted of a single-mode fiber (SMF) no core fiber (NCF) single-mode fiber structure (shortened as SNS structure) with a large-lateral offset spliced between the two sections of multimode fiber (MMF). The sensitivity was up to -13703.63 nm/RIU and -13160 nm/RIU in the refractive index range of 1.3370 to 1.3410 based on the shift of the interference spectrum. Moreover, the sensor showed a good linear response and high stability, with an RSD of only 0.91% for the trough of the interference in air over 1 h. The paper is well written. Both theoretical and experimental contents are satisfactory. In short, I think the content of this paper is ready for publication.

Reviewer 2 Report

Comments and Suggestions for Authors

In this paper, the authors present a fiber sensor for the refractive index sensing of seawater based on a Mach-Zehnder interferometer has been demonstrated. The sensor consisted of a single-mode fiber (SMF) - no core fiber (NCF) - single-mode fiber structure (shortened as SNS structure) with a large-lateral offset spliced between the two sections of multimode fiber (MMF). The optimization studies of the multi-mode fiber length offset SNS length, and vertical axial offset distance have been performed to improve the coupling efficiency of interference light and achieve the best extinction ratio. In the experiment, the large-lateral offset sensor was prepared to detect the refractive index of various ratios of saltwater, which were used to simulate the seawater environment. This article is clear, concise, and suitable for the scope of the journal. Several suggestions are supplied:

1. Suggest the authors supply the length label for pic in Fig.7.

2. Suggest the authors supply more detail about the stability performance of the device.

3. Suggest the authors supply more detail in sentences about the comparison of sensor performance between the previous and proposed sensor.

4. Optical fiber sensing for the marine environment and marine structural health monitoring a review is an emerging area, suggest supplying more review works on this topic in the introduction part. 10.1016/j.optlastec.2021.107082

Reviewer 3 Report

Comments and Suggestions for Authors

This paper presents a sensor for salinity measurements (by RI sensing) based on a  large-lateral offset Mach-Zehnder Interferometer. Authors state the novelty, however I do not see it. This design (Mach-Zehnder configuration) was already proposed in previous contributions, such as: 10.1016/j.sna.2018.08.026 - NCF design already presented, 10.1080/00150193.2022.2079453, even with combining different fiber types in-line (10.1016/j.infrared.2022.104134). Those papers were not even cited by the authors in this paper. Therefore, also the state of the art was not done consciously.

For this reason I cannot recomend publishing this paper.

Reviewer 4 Report

Comments and Suggestions for Authors

Dear Authors,

The article entitled “Highly Sensitive Determination of Refractive Index of Seawater Based on Mach-Zehnder Interferometer with Large Lateral Offset” aims to present a fiber sensor with a novel structure. Despite the promising topic and potential value of the research, the article suffers from a several issues that need to be addressed before it can be considered for publication. Based on my review of your article, I have the following comments and suggestions for improvement:

1) Figure 2: On Figure 2, it is necessary to indicate the geometric arrangement of the optical fiber parts. In addition, the figure does not indicate the parameters of the selected fiber. The chosen scale on the Z-axis does not allow for a proper evaluation of the mode structure for your sensor, especially for elements with sizes on the order of 100 µm. You need to zoom in by a factor of at least 10. The figure should also show not only the longitudinal but also the transverse distribution of the field in the fiber. In Figure 2b the Monitor Value starts at 1.0 and goes down to almost zero, does this mean that almost all of the light power is lost? A more detailed explanation of the Monitor Value graph and its relationship to sensor performance is needed.

2) Figure 3: The right y-axis label in Figure 3a should be corrected from “ER (nm)” to “ER (dB)”. It is also unclear why a step size of 50 µm was chosen in Figure 3. The function seems to change rather abruptly, and it is possible that the maximum ER is achieved between 50 and 100 µm SMF. In addition, it is not clear why a NCF length of 200 µm was chosen for the experiment and not 100 µm where, according to the plot in Figure 3c, the sensitivity is higher.

3) Figure 10: Figure 10b has 9 experimental points corresponding to dips, but Figure 10a shows only 7 experimental spectrum plots. The missing spectrum plots should be added.

4) Table 1: In Table 1, the comparison is made with sensor sensitivities an order of magnitude lower than in your paper. However, there are known works with comparable sensitivities [1] ~ 13000 nm/RIU, [2] ~ 14000 nm/RIU, [3] ~ 15000 nm/RIU. You should cite these works in the table and compare the advantages and disadvantages of your implementation with fiber sensors of similar sensitivity. In the discussion section, it is appropriate to discuss aspects such as potential limitations, practical application problems, and future work.

5) Finally, correct the text “(in the range of 6 to 40)” by adding “µm” for clarity in the measurement units.

6) I also recommend adding a link to a recent review article [4] on seawater sensors. 

[1] Gao, S.; Zhang, W.; Bai, Z.; Zhang, H.; Geng, P.; Lin, W.; Li, J. Ultrasensitive Refractive Index Sensor Based on Microfiber-Assisted U-Shape Cavity. IEEE Photonics Technology Letters 2013, 25, 1815–1818, doi:10.1109/LPT.2013.2274492.

[2] Wang, B.-T.; Wang, Q. An Interferometric Optical Fiber Biosensor with High Sensitivity for IgG/Anti-IgG Immunosensing. Optics Communications 2018, 426, 388–394, doi:10.1016/j.optcom.2018.05.058.

[3] Zheng, H.; Lv, R.; Zhao, Y.; Tong, R.; Lin, Z.; Wang, X.; Zhou, Y.; Zhao, Q. Multifunctional Optical Fiber Sensor for Simultaneous Measurement of Temperature and Salinity. Opt. Lett., OL 2020, 45, 6631–6634, doi:10.1364/OL.409233.

[4] Li, G.; Wang, Y.; Shi, A.; Liu, Y.; Li, F. Review of Seawater Fiber Optic Salinity Sensors Based on the Refractive Index Detection Principle. Sensors 2023, 23, 2187, doi:10.3390/s23042187.

  Comments on the Quality of English Language

Dear Authors,

I have noticed grammatical errors in your manuscript. Here are a few examples:

- "The sensor was consisted of" should be "The sensor consisted of"

- "These sensors are typically susceptible to damage and have a short lifespan due to the reduce of the diameter of the fiber" should be "reduced"

- "The optimization studies of the multimode fiber length, offset SNS length, and vertical axial offset distance have been performed to improve the coupling efficiency..." should be "were performed"

These are just a few examples of grammatical errors. I recommend that you carefully check your manuscript for correct use of tense.

Reviewer 5 Report

Comments and Suggestions for Authors

In this work, the authors demonstrate a SM-OSNS fiber sensor for RI detection of seawater. Design optimizations have been done on the length of the multimode fiber, the length of the offset SNS, and the vertical axial offset distance. The experimental results show good linear response within the RI range of 1.337 - 1.341. The sensitivity is reported to be -13703.63 nm/RIU by linear fitting. The manuscript has been properly organized and presented. Before it can be accepted, following issues are to be addressed.

1.     For the seawater detection, does the author consider the practical scenario that there are many components in the water. What will these impurities affect the test?

2.     As the authors have mentioned, the structure of proposed sensor is very fragile, even considering the possible packaging. Therefore, could the authors give a practical suggestion to overcome this issue?

Comments on the Quality of English Language

1.  There are still some syntax errors in the manuscript, such as in Line 295 “In conclusion, an SM-OSNS fiber sensor”, in fact, it should be “a SM-OSNS…”

2.  The abbreviation firstly appears should be noted clearly, for example, in Line 20, no explanation for the “RSD”
